# Natural Antibacterial Reagents (*Centella*, Propolis, and Hinokitiol) Loaded into Poly[(*R*)-3-hydroxybutyrate-*co*-(*R*)-3-hydroxyhexanoate] Composite Nanofibers for Biomedical Applications

**DOI:** 10.3390/nano9121665

**Published:** 2019-11-22

**Authors:** Rina Afiani Rebia, Nurul Shaheera binti Sadon, Toshihisa Tanaka

**Affiliations:** 1Interdisciplinary Graduate School of Science and Technology, Shinshu University, 3-15-1 Tokida, Ueda, Nagano 386-8567, Japan; rina_rebia@yahoo.com; 2Faculty of Textile Science and Technology, Shinshu University, 3-15-1 Tokida, Ueda, Nagano 386-8567, Japan

**Keywords:** antibacterial effect, centella, propolis, hinokitiol, biodegradable polymer, PHBH, nanofiber

## Abstract

*Centella asiatica*, propolis, and hinokitiol, as natural antibacterial reagents, were integrated into the poly[(*R*)-3-hydroxybutyrate-*co*-(*R*)-3-hydroxyhexanoate] (PHBH) polymer to produce antibacterial wound dressings, using electrospinning process. The results showed that the fiber diameters and surface morphology of PHBH composite nanofibers were influenced by the addition of ethanol–*centella* (EC), methanol–*centella* (MC), ethanol–propolis (EP), and ethanol–hinokitiol (EH) at various ratios compared to pristine PHBH nanofibers. From FT-IR, the nanofibrous samples with higher contents of natural antibacterial substances showed the peaks of carboxylic acid, aromatic ring, and tropolone carbon ring from *centella*, propolis, and hinokitiol, respectively. Furthermore, the tensile strength of neat PHBH nanofibers was increased from 8.00 ± 0.71 MPa up to 16.35 ± 1.78 MPa by loading of propolis (EP) 7% into PHBH. X-ray analysis explained that the loading of propolis (EP) was also able to increase the crystallinity in PHBH composite nanofibers from 47.0% to 54.5%. The antibacterial results demonstrated that PHBH composite nanofibers containing natural antibacterial products were potent inhibitors against the growth of *Escherichia coli* and *Staphylococcus aureus*, amongst them hinokitiol and propolis proved to be the most effective. Additionally, the release studies displayed that *centella* and hinokitiol had faster release from PHBH composite nanofibers in comparison to propolis.

## 1. Introduction

Electrospinning is a straightforward and efficient method to fabricate nonwoven mats with continuous fibers in the range from micrometer down to nanometer. Nowadays, nanofibers produced by electrospinning have received the considerably high attention to be an excellent candidate for many important applications in medical field such as drug delivery [1], oral cavity [2], tissue engineering [3,4,5,6], wound dressing and healing [7,8] due to the ultra-fine diameter, high surface area to volume ratio, and cost-effectiveness. A large number of studies devoted to utilizing nanofibers that could mimic the extracellular matrix (ECM) of the body and cellular activity [9], as well as sustained drug action such as in skin regeneration or healing [1,10,11] have been published over the last few years. Open wounds were reported to be healed in a much faster-recovering pace by covering from infection using antimicrobials. Without covering, the healing process could be hindered by bacterial interference with cell-matrix interactions, followed by the delay of the cell proliferation and tissue regeneration [12]. Recent studies have endeavored to incorporate novel drugs, antibacterial reagents, and other healing enhancers into nanofiber membranes for the sake of facilitating the wound healing process [7,10,11].

Natural antibiotics as one of herb, plant extracts, or pure compounds have been used in pharmaceutical products for centuries. With the advancement of nanotechnology, it provides unlimited opportunities for employing natural antibiotics as additives for engineering novel intriguing materials for wound treatments and healing. One of the advantages of using natural products such as plant extracts, essential oils, aloe vera, honey, and so on instead of synthetic reagents (iodine and silver) is to diminish the risk of sensitization and development of resistance [12]. *Centella asiatica*, a traditional medicine derived from plants, has been used widely in Southeast Asia, India, and other regions for skin treatments. The major biologically active compounds in *centella* are triterpenoids, which include asiaticoside, madecassoside, asiatic acid, and madecassic acid [13,14,15]. The previous study of gelatin/*centella asiatica* [14] reported that *centella* extract demonstrated a strong inhibition against microbial pathogens such as *S. aureus*, *E. coli*, and *P. aeruginosa*, which are most commonly implicated in wound infection. Polycaprolactone/*centella* biopolymer nanofibers also showed the antibacterial activity against *B. cereus* and *M. luteus* with the improvement of mechanical properties owing to the decrease of the fiber diameters after adding *centella* [5]. *In vitro* test using human dermal fibroblast and human dermal keratinocyte (HaCaT) cells and *in vivo* circular wound excision of rabbits, *centella* extract proved to accelerate wound healing process and the cell migration rates [16]. Furthermore, electrospun gelatin membrane containing *centella* improved cell proliferation *in vitro* for fibroblasts (L-929) and wound recovery *in vivo*, using male Sprague Dawley (SD) rats [14].

Propolis, a resinous substance with dark brown color, was produced via mixing collected from buds and bark of trees with beeswax and saliva by honeybees. The main chemical compounds in propolis are aromatic acids and phenolic compounds, especially flavonoid and phenolic acid [17,18,19]. A number of studies have demonstrated that propolis exhibited great potential for wound healing. In the study of polyurethane/propolis nanofibers [20], the authors concluded that nanofiber mats containing 30% (*w*/*w*) propolis had strong inhibitory effects against *E. coli*. In addition, PVA/hydroalcoholic extract propolis nanofiber [21] exhibited the antibacterial activity almost equal to the vancomycin, an effective commercial antibiotic. Propolis was also confirmed to possess antibacterial, antifungal, antiviral, antioxidant, anti-inflammatory, and antitumor characteristics [22].

Hinokitiol is one of the natural components isolated from Japanese cypress and western red cedar. It belongs to the class of tropolones that contain an unsaturated seven-membered carbon ring with *β*-Thujaplicin and is well-known for its inhibitory effects against the growth of fungi, bacteria, and insects [23]. Research on hinokitiol as a potential medication for the treatment of dental root canal demonstrated strong antibacterial effects against *S. aureus* and anti-inflammation [24]. These natural products (*centella*, propolis, and hinokitiol) show the capability to be used as alternative antibacterial reagents to replace conventional synthetic antiseptics.

Poly(3-hydroxybutyrate-*co*-3-hydroxyhexanoate) (PHBH) is a copolymer, developed from poly[(*R*)-3-hydroxybutyrate] (PHB), the basic homopolymer in Polyhydroxyalkanoates (PHA), which are a family of polyesters produced by bacteria through the fermentation of sugars, lipids, fatty acids, alkanes, alkenes, and alkanoic acids [1,6,25,26,27]. PHBH, one of copolymers in more than 150 kinds of PHAs, which in turn consist of various co-monomers, has been considered for commercial production [28]. In the biomedical field, PHB and its copolymers have been demonstrated to be fully biodegradable and an excellent biological material with biocompatibility properties. The prominent advantage of these polymers is their ability to be completely degraded by microorganisms under aerobic and anaerobic conditions without forming toxic by-products [26,29]. Recently, these materials have been developed for medical applications such as suture, skin scaffold, tissue engineering, and so on. Moreover, biodegradable nanofibers as a material matrix for wound dressing and healing have received considerable attention because they have been found to be an effective strategy for delivering antibacterial agents or anti-inflammatory substances. Previous studies demonstrated that PHB/caffeic acid electrospun nanofibers coated with polyelectrolyte exerted the antibacterial activity and antitumor [7]. Poly(3-hydroxybutyrate-*co*-3-hydroxyvalerate) (PHBV)/Curcumin nanofibers showed positive effects on wound healing, cell adhesion, and non-toxicity to L929 mouse fibroblast cell line [11]. Besides, PHBH nanofibers promoted the NIH3T3 fibroblast cell attachment and proliferation [4]. This polymer can be regarded as one of the most promising polymers obtained from renewable resources, suitable for drug or antibacterial products carrying.

In this study, *centella*, propolis, and hinokitiol with inherent biological activity were selected as representatives for natural antibacterial products and were integrated in PHBH biodegradable polymer. We focused on developing nanofibrous membranes with antibacterial properties, comprised of natural antibacterial reagents and biodegradable PHBH polymer for combined effects on wound healing. The electrospinning technique was chosen as the mean of fabrication. To our knowledge, the fabrication and characterizations of *centella*, propolis, and hinokitiol loaded in biodegradable PHBH polymer have not been reported hitherto. We attempted to investigate the effects of the addition of *centella*, propolis, and hinokitiol to PHB copolymer with 3-hydroxyhexanoate (3HH) unit as the polymer matrix regarding crystallinity of the PHBH composite nanofibers, antibacterial activities, and sustained release behaviors. The advantages of this research are to promote the use of natural products over conventional synthetic antiseptics by studying three variants of natural antibacterial products, namely *centella*, propolis, and hinokitiol. It is also important to present harmlessness for products made of PHBH and natural antibacterial reagents if they are to be used in medical applications.

The surface morphology, chemical characterization, mechanical properties, crystallinity of PHBH, zone of inhibition, and sustained release behaviors of *centella*, propolis, and hinokitiol loaded into PHBH were analyzed by scanning electron microscopy (SEM), Fourier Transform Infrared (FTIR) Spectroscopy, tensile testing machine, X-ray diffraction, antibacterial experiment, and UV-visible spectrophotometer, respectively. From these results, the influences of natural products and solvents to PHBH structure and the properties of the resultant nanofibrous composites will be revealed.

## 2. Materials and Methods

### 2.1. Materials

PHBH containing 5.5 mol% of 3HH (*M*_w_ = 5.37 × 10^5^ g/mol%) was provided by Kaneka Corp., Osaka, Japan. The natural antibacterial reagents, *centella asiatica* powder, propolis powder, hinokitiol powder were purchased from Dins Natural (pvt) Ltd., Matara, Sri Lanka; Stakich, Inc., Troy, USA; and Wako Pure Chemical Industries, Ltd., Osaka, Japan, respectively. Solvents, used to prepare electrospinning solutions, were 1,1,1,3,3,3-hexafluoro-2-propanol (HFIP) purchased from Wako Pure Chemical Industries, Ltd., Japan. Acetone, ethanol 99.5%, and methanol were obtained from Wako Pure Chemical Industries, Ltd., Japan.

*Staphylococcus aureus* (NBRC 12732) and *Escherichia coli* (NBRC 3972) for antibacterial activity tests were purchased from the National Institute of Technology and Evaluation, Biological Resource Center (NBRC), Tokyo, Japan.

### 2.2. Electrospinning Process of PHBH Composite Nanofibers Containing the Natural Antibacterial Reagents

#### 2.2.1. Preparation of Natural Antibacterial Reagent Solutions

*Centella* solutions were prepared using either ethanol or methanol the concentrations of *centella* to solutions were 15% or 30% (*w*/*v*), respectively. The propolis solutions were prepared by dissolving propolis in acetone at 10% (*w*/*v*) or ethanol with the concentrations of 10% or 30% (*w*/*v*), respectively. Each prepared solution was mixed using a magnetic stirrer for 24 h at room temperature. Afterwards, all solutions were filtered to remove insoluble substances. The hinokitiol solution was prepared by dissolving hinokitiol powder in 99.5% ethanol with a concentration of 30% (*w*/*v*), also stirred at room temperature for 24 h without filtration.

The concentrations of natural products in ethanol–*centella* (15EC and 30EC), methanol–*centella* (15MC and 30MC), acetone–propolis (10AP), and ethanol–propolis (10EP and 30EP) were determined from the dry weight of each natural product by evaporating solutions under vacuum. Ethanol–hinokitiol (30EH) mentioned the same as 30% because of no insoluble substances in the ethanol solution.

These percentage concentrations of *centella*, propolis, hinokitiol are displayed in Table 1. These solutions were added to PHBH HFIP solutions with different ratios before the electrospinning process (The ratios will be explained in the next step of preparation of PHBH with natural product solutions).

#### 2.2.2. Preparation of PHBH with Natural Antibacterial Reagent Solutions for Electrospinning

PHBH was dissolved in HFIP at room temperature for 4 h to make solution 2 wt%. The natural antibacterial solutions were added into PHBH HFIP solutions and stirred continuously for 16–20 h to get homogeneous solutions. The ratios of ethanol–*centella* (15EC and 30EC) or methanol–*centella* (15MC and 30MC) were 1%, 5%, and 10% over the total weight of spinning PHBH solutions. Furthermore, ratios of acetone-propolis (10AP), ethanol–propolis (10EP and 30EP), and ethanol–hinokitiol (EH) to total weight of spinning PHBH solutions were 1%, 5%, and 7%, Table 1. These prepared blend solutions were contained into a glass syringe (10 mL) with 0.6 mm diameter of stainless steel needle. The electrospinning process was carried out using a high voltage power supply by Kato Tech, Japan. The solution feeding rate was 0.1–0.3 mm/min, 15–20 kV voltage, and 15 cm tip-to-collector distance. The environmental conditions of the electrospinning chamber were at a humidity level of 20–30 RH% and temperature ranged between 23–26 °C. The electrospun fibers were collected on the surface of the collector followed by drying at room temperature for 24 h to evaporate the solvents.

### 2.3. Characterization of PHBH Composite Nanofibers with Natural Antibacterial Reagent 

#### 2.3.1. Scanning Electron Microscopy (SEM)

The morphology of electrospun nanofibers was investigated by scanning electron microscopy (SEM, JSM-6010LA, JOEL, Tokyo, Japan) at an accelerating voltage of 10 kV with various magnifications. The samples were coated with platinum (Pt) in a sputtering device for 60 s at 30 mA. Image J software was used to measure 50 fibers diameters from each SEM photograph.

#### 2.3.2. Fourier Transform Infrared (FT-IR) Spectroscopy

The presence of *centella*, propolis, and hinokitiol incorporated in electrospun nanofibers were analyzed using attenuated total reflectance (ATR) FT-IR spectroscopy (IR Prestige-21, Shimadzu Corp., Kyoto, Japan). The infrared spectra were recorded from 4000–600 cm^−1^, with a resolution of 4 cm^−1^. The infrared beam enters the ATR crystal at an angle of typically 45° and is totally reflected at the crystal to sample interface with a low signal of noise ratio.

#### 2.3.3. X-ray Diffraction

Wide-angle X-ray diffraction (WAXD) was used for crystalline structure analysis of PHBH nanofibers with or without *centella*, propolis, or hinokitiol. The two-dimensional (2D) patterns of nanofibers were recorded by X-ray diffraction equipment (SPring-8 synchrotron radiation facility in Japan) with a wavelength of 0.07085 nm at a 2θ scanning angle between 5–30°. The distance from the sample to the detector (PILATUS 3 × 2 M) was 326.5 mm and the exposure time was 2.0 s. The X-ray diffraction of composite nanofibers was investigated by the MDIP application.

#### 2.3.4. Mechanical Properties

The mechanical properties of neat PHBH, PHBH/*centella*, PHBH/propolis, PHBH/hinokitiol nanofibers were investigated by using a tensile testing machine (EZTest/EZ-S, Shimadzu Corp., Kyoto, Japan) with 50-N load cell at room temperature. Ten specimens were prepared from each composite nanofiber sheet. The specimens with thickness ranging from 0.05 to 0.14 mm were cut into a rectangular shape of 30 mm × 5 mm. The ends of each sample were secured by two clamps with an initial distance of 10 mm and a crosshead speed of 10 mm/min. Tensile strength, Young’s modulus, and elongation at break were calculated on the basis of the stress-strain curves.

### 2.4. Sustained Release Behavior of Natural Antibacterial Reagent from PHBH Composite Nanofibers

The release of *centella*, propolis, and hinokitiol was analyzed by a UV-visible spectrophotometer (UV2550, Shimadzu Corp., Kyoto, Japan). For release experiments, PHBH composite nanofibers with *centella*, propolis, or hinokitiol were cut into 2 × 2 cm and measured specimen weight. For each sample, 2 specimens were placed in a test tube filled with 50 mL of phosphate-buffered saline (PBS) (pH 7.4) incubated at 37 °C and stirred at 150 rpm. During the release, 4 mL of the supernatant was retrieved from the release medium at the time points of 5, 10, 20, 30, 40, 50, 60, 120, 240, 360, 480, 600, 720, 1440, 2880, 4320, and 5760 min and the same amount of fresh PBS was added immediately to maintain the medium volume.

The solutions in various concentrations were used to make a standard calibration curve at wavelength 200–700 nm. In detail, certain weight of extract *centella*, extract propolis, hinokitiol (0.8 mg) was dissolved in 100 mL ethanol/PBS or methanol/PBS (1:99, v/v) and then diluted to 400, 200, 100, 50, 25, 12.5, 6.25, 3.125, and 1.56 μg/mL. UV-vis spectrophotometer at a wavelength of 270 nm, 273 nm, 224 nm was used to measure the amount of released *centella*, propolis, and hinokitiol from PHBH composite nanofibers, respectively. The total amounts of *centella*, propolis, and hinokitiol from the nanofiber mats were determined as the average value of the three tests. These results were presented in the form of a cumulative amount of release [30]:
Cumulative amount of release (%) = *M_t_*/*M*_∞_ × 100,
where *M_t_* is the amount of *centella*, propolis, or hinokitiol released at the time *t*, *M*_∞_ is the total amount of *centella*, propolis, and hinokitiol loaded in PHBH composite nanofibers.

### 2.5. Antibacterial Activity Test

To examine the antibacterial activity of natural antibacterial reagents, *S. aureus* and *E. coli* were chosen as representatives for gram-positive and gram-negative bacteria. The test method was the disk diffusion test, which has been well established by previous reports [2,20,30,31,32,33]. The bacterial culture was spread on the Luria Bertani (LB) agar surface by using a sterile cotton bat. The PHBH/*centella*, PHBH/propolis, and PHBH/hinokitiol composite nanofibers were placed on the surface of the Petri dishes then they were incubated at 30 °C (*E. coli*) and 37 °C (*S. aureus*) for 24 h. The diameters of inhibition zone (mm) were measured for 3 specimens each sample and the results were expressed as mean diameters with standard deviations (in millimeters).

### 2.6. Statistical Analysis

All experiments were conducted in triplicate and the data are presented as mean ± standard deviation (SD). The significant differences were statistically analyzed by one-way analysis of variance (ANOVA) using R free software. Statistical significance was set at *p* < 0.05 to identify which groups were significantly different from other groups.

## 3. Results and Discussion

### 3.1. Morphology of PHBH Composite Nanofibers with Natural Antibacterial Reagents

The surface morphology of PHBH nanofibers containing different natural antibacterial products was investigated using SEM. Representative SEM images and the fiber diameter distributions of neat PHBH and PHBH nanofibers loaded with different amounts of *centella* solutions (1%, 5%, and 10%), propolis solutions (1%, 5%, and 7%), and hinokitiol solutions (1%, 5%, and 7%) are exhibited in Figure 1. In general, the different natural product solutions had effects on the surface morphology of composite nanofibers.

Table 2 shows the percentage of each natural product solution used, the conditions of the electrospinning process, the average diameters, standard deviations (SD), maximum, and minimum fiber diameters of each sample. From those data, it can be confirmed that PHBH/ethanol–*centella* (15EC and 30EC) and PHBH/methanol–*centella* (15MC and 30MC) composite nanofibers showed the uniform fiber diameter. These also exhibited the same trend of a decrease in fiber diameter by increasing the ratio of *centella* solutions (EC and MC) in spinning PHBH solution. The fiber diameter, with high concentrations of *centella* solution in PHBH/30EC (10%) and PHBH/30MC (10%), diminished to 349 ± 108 nm and 332 ± 62 nm, respectively. This phenomenon may be due to the decrease in the viscosity of polymer solutions by the addition of ethanol or methanol into HFIP. The viscosity of the mixture solution reduced when the considerable amounts of *centella* solution added into the polymer matrix [5,14]. Interestingly, the high concentration of *centella* in PHBH/30EC (10%) (Figure 1) resulted in fibers merging.

Such a bonded structure was also appeared in PHBH/30EP (7%). The bonded structure in high concentration solutions of EC and EP might be due to using ethanol as the solvent and mixing with HFIP. By contrast, PHBH/30EP (7%) was accompanied by an increase in fiber diameter by loading a high concentration of 30EP. The fiber diameter was increased from 539 ± 99 nm PHBH/30EP (1%) to 739 ± 197 nm PHBH/30EP (7%). A similar behavior was obtained from cellulose acetate nanofibers with a high concentration of honey bee propolis prepared by ethanol solution [33]. Kim et al. [20] also reported that the propolis concentration increased the fiber diameter and provided the linking of PU fibers due to its adhesive properties and demonstrated the optimum utilization of the bonding element for reinforcement of nonwoven fabric. However, different results of fiber diameter were obtained from PHBH composite nanofibers containing acetone-propolis (PHBH/10AP), which are 520 ± 83 nm, 527 ± 161 nm, and 529 ± 109 nm in the cases of 1%, 5%, and 10% (*v*/*v*) of 10AP, respectively. This is probably because the evaporation of acetone was faster than ethanol during the electrospinning process.

For PHBH/ethanol–hinokitiol (EH) composite nanofibers, the surface morphology of PHBH/30EH (1%) was smooth, homogeneous without beads. The average diameter was 562 ± 87 nm with a majority of fibers in the range of 438 nm to 745 nm. However, it was difficult to obtain uniform nanofibers with high ratios of hinokitiol solutions (EH), more than 1%. As shown in Figure 1 of PHBH/30EH (5%) and (7%), these resulted in aligned fibers with small grains like beads.

The final surface morphologies of PHBH composite nanofibers were influenced by the kind of used natural products and concentration ratios of natural product solutions to PHBH solutions. This result clearly indicates that the fiber diameters and their distributions can be controlled by concentration ratios of natural products in solutions. In this paper, PHBH composite nanofibers with natural product solutions of PHBH/30EC (10%), PHBH/30MC (10%), PHBH/30EP (7%) and PHBH/30EH (1%) were chosen for further investigation by FTIR, WAXD, mechanical properties, antibacterial test, and release characteristic because these samples had fiber diameter between 300–1000 nm, surface morphology without beads, and high concentration of natural products.

### 3.2. Characteristic of PHBH Composite Nanofibers with Natural Antibacterial Product 

#### 3.2.1. FT-IR Spectral Analysis

The functional groups of natural antibacterial products contained in PHBH composite nanofibers were analyzed by using ATR (FT-IR). The FT-IR spectra of PHBH/30EC (10%), PHBH/30MC (10%), PHBH/30EP (7%), and PHBH/30EH (1%) are shown in Figure 2. The FT-IR spectra of neat PHBH nanofiber and PHBH composite nanofibers with natural products showed similar peaks. The characteristic peaks of PHBH at 2970, 2937, 2870, and 1719 cm^−1^ were attributed to the stretching vibrations of C–CH_3_, CH_2_, CH, and C=O, respectively [4].

FT-IR graphs of *centella* powder from ethanol and methanol solutions after filtering and drying exhibit some bands centered at 3320 or 3300 cm^−1^ corresponding to O–H stretching vibration of carboxylic acid, whereas, 1690 or 1638 cm^−1^ peaks characterizing C–O stretching vibration. The bands at 1459, 1380, and 1025 cm^−1^ shows the presence of alkenes with C–H in-plane bending, the stretching vibration of C–N for aromatic amide group, and C–O stretching, respectively (Figure 2A). These spectra of those compounds were confirmed in the research of *centella asiatica* by Manotham et al. [5], Sondari et al. [34], and Sugunabai et al. [35]. As showed in Figure 2A, it proved that using methanol as solvent yielded higher content of Asiatic acid than using ethanol with the same concentration from raw *centella* (30%EC and 30%MC). In the case of PHBH/30MC (10%) composite nanofibers, the decline of the band at 3340 cm^−1^ suggested the formation of the intermolecular hydrogen bond between *centella* and PHBH polymer.

The FT-IR spectra of propolis powder and PHBH/30EP (7%) nanofibers were exhibited in Figure 2B. In the case of propolis powder, the band at 3340 cm^−1^ was assigned to stretching vibration of the O–H group in the phenolic compound and the band at 2920–2870 cm^−1^ was attributed to C–H aliphatic stretching vibration (stretching vibration of CH_2_ and CH_3_) [36]. The bands at 1706 cm^−1^, 1650–1602 cm^−1^, and 1190 cm^−1^ were attributed to the C=O group, aromatic ring deformations C=C stretching vibration, and C–O stretching vibration, respectively. In PHBH/30EP (7%) composite nanofibers, some peaks appeared at between 1651 and 1610 cm^−1^, and 1510 cm^−1^, confirming the presence of propolis in PHBH nanofiber mats. These bands were assigned to stretching of C=C aromatic ring bands in flavonoids [17,36].

The characteristic absorption peaks of hinokitiol powder were observed at 3200, 1609, 1543, 1476, 1459, 1417, 1269, 1185, and 950 cm^−1^. These bands represented O–H stretching, C=C and C=O stretching, C=C stretching (in-phase), ring CH bending, C–O–H group, and ring CH bending, respectively [37]. Dyrskov et al. [23] reported that the absorbance bands in the hinokitol spectrum ascribed to C=O stretching in the tropolone carbon ring (1609 and 1543 cm^−1^) and the C–O–H group (1189 cm^−1^). The presence of hinokitiol in PHBH composite nanofibers was confirmed by the weak peak appeared at 1610 cm^−1^ that is related to tropolone carbon ring. In general, the peak intensity decreased by loading natural product might reveal that natural product is homogeneously distributed through the polymer matrix [20]. The nanofibrous samples with higher contents of natural antibacterial substances showed the peaks of carboxylic acid, aromatic ring, and tropolone carbon ring from *centella*, propolis, and hinokitiol, respectively.

#### 3.2.2. Crystalline Structure by Wide-Angle X-ray Diffraction (WAXD)

The WAXD measurement was performed in order to confirm the influence of loading *centella*, propolis, and hinokitiol on the crystallinity of PHBH. All reflections of WAXD and intensity profile are displayed in Figure 3. The neat PHBH nanofibers exhibit diffraction peaks at 2θ = 6.1°, 7.7°, 9°, 10°, 11.7°, 12.4°, and 13.8° which were assigned to the (020), (110), (101), (121), (040), and (002) of the orthorhombic unit cell of PHB crystal, respectively [38]. The intensity of diffraction peaks of neat PHBH nanofibers slightly increased by loading *centella* and propolis in PHBH/30EC (10%) and PHBH/30EP (7%) composite nanofibers. Moreover, the calculated crystallinity of neat PHBH (47.0%) was slightly lower than that of PHBH/30EC (55.0%), PHBH/30MC (49.2%), and PHBH/30EP (54.5%) in Table 3. It was previously reported that the crystallinity of PHB nanofibers increased by adding the natural phenolic compound caffeic acid [30]. Kim et al. [39] supposed that intermolecular interaction through a hydrogen bonding of plant polyphenol with polycaprolactone probably increased by slight increment of crystallinity in polycaprolactone. Whereas, the crystallinity of PHBH/30EH (1%) composite nanofibers was declined to 44.1% compared to neat PHBH. These results indicate that an intermolecular interaction between PHBH polymer and *centella* or propolis in nanofibers mat exists.

#### 3.2.3. Mechanical Characteristic of Composite Nanofibers

Tensile test was conducted on neat PHBH and PHBH composite nanofibers with the natural antibacterial products in order to evaluate the influences of natural product loading. The tensile strength, elongation at break, and Young’s modulus of neat PHBH, PHBH/30EC (10%), PHBH/30MC (10%), PHBH/30EP (7%), and PHBH/30EH (1%) are shown in Table 3. Figure 4 illustrates the representative stress–strain curves of PHBH composite nanofiber samples. The tensile strength of neat PHBH was 8.0 MPa with high elongation up to 61.5%. Interestingly, the loading of *centella* (EC 10%) and propolis (EP 7%) in PHBH demonstrated the same trend with the tensile strength rose up to 17.8 and 16.4 MPa but it reduced elongation down to 15.6% and 9.3%, respectively. PHBH/30MC (10%) composite nanofibers exhibited no significant differences in tensile strength and Young’s modulus compared to neat PHBH. PHBH/30EH (1%) resulted in a decline in both tensile strength and elongation compared to neat PHBH.

PHBH composite nanofibers with *centella* (EC) and propolis (EP) using ethanol as solvent displayed higher tensile strength in comparison to neat PHBH, PHBH/MC, and PHBH/EH. These phenomena may be due to the surface morphology and polymer structure in PHBH composite nanofibers. The surface morphology of PHBH/30EC (10%) and PHBH/30EP (7%) showed bonded-structure between fibers, whereas PHBH/30MC (10%) and PHBH/30EH (1%) surface revealed randomly oriented fibers. Kim et al. [20] reported the similar behaviors of merged structure, the adhesive properties of propolis might be useful to increase the mechanical strength of polyurethane fibers. The mechanical properties of nonwoven nanofibers also depended on parameters such as surface interaction among the fibers, average fiber diameter, fiber defects (bead formation) during electrospinning [10,11].

Additionally, the enhancement of physical properties in PHBH/30EC (10%) and PHBH/30EP (7%) might be related to the rise of the crystallinity of PHBH polymer in PHBH composite nanofibers. It is probably due to the interaction of natural products (EC and EP) to PHBH polymer that was explained by FT-IR and X-ray analysis.

### 3.3. In Vitro Antibacterial Activity

The antibacterial activity of neat PHBH nanofibers, PHBH composite nanofibers with *centella*, propolis, and hinokitol against the gram-positive bacteria *S. aureus* and gram-negative bacteria *E. coli* was evaluated using inhibition zone method [20,33,40,41]. The antimicrobial effect of samples was evaluated at 37 °C (*S. aureus*) and 30 °C (*E. coli*) for 24 h. Table 4 presents the diameters of the inhibition zone for PHBH composite nanofibers with natural products. The effects of natural antibacterial products via the agar diffusion test are clearly shown in Figure 5. PHBH/30EC (10%) was observed to have no effect against gram-positive and gram-negative bacteria. Whereas, *centella* in PHBH/30MC (10%) demonstrated low antibacterial activity against *S. aureus* with inhibitory zone of 7.7 mm and no effect against *E. coli*. The diameter of the inhibition zone of PHBH/30EP (7%) was 18.3 mm for *S. aureus* and 17.3 mm for *E. coli*. PHBH/30EH (1%) composite nanofibers have extensive inhibition zone against *S. aureus* (25.7 mm) and *E. coli* (29.7 mm), respectively. The zones of inhibition for PHBH/30EP (7%) against *S. aureus* were slightly broader than that against *E. coli*. Zeighampour et al. [21] explained that hydroalchoholic extract of propolis showed better antibacterial activities against gram-negative bacteria than gram-positive bacteria due to the different cell wall chemical structures of bacteria. However, the opposite results for PHBH/30EH (1%) might be due to the different modes of action and the bactericidal effects of natural products.

Regarding these results, the antimicrobial effects of PHBH composite nanofibers with propolis and hinokitiol were observed to be better than these of *centella*. However, the low effect antibacterial activity of *centella* (EC or MC) might be due to the low chemical content in PHBH composite nanofiber as analysed by FT-IR, which PHBH/30EC (10%) showed no clear detection of asiatic acid. For further work, raising the *centella* concentrations in ethanol or methanol solutions might be one of the solutions to increase the antibacterial activity, if those will be used for wound healing application. Overall, the natural products (especially propolis and hinokitiol) loaded into PHBH composite nanofibers inhibited the growth of *S. aureus* and *E. coli* powerfully.

### 3.4. Release Behavior of Natural Antibacterial Product

The release profiles of *centella*, propolis, and hinokitiol from electrospun PHBH composite nanofibers were plotted as a function of time in PBS with pH 7.4 at 37 °C. The release curves of PHBH/30EC (10%), PHBH/30MC (10%), PHBH/30EP (7%), and PHBH/EH (1%) composite nanofibers demonstrated different release behaviors in each sample as shown in Figure 6. *Centella* from PHBH/30EC started to release within the 10 min and completely released in 20 min. In the case of PHBH/30MC, Figure 6B, the similar behaviors were also observed with complete release in 20 min. The maximum releases of *centella* from PHBH/30EC and PHBH/30MC were 13.9 % and 31.5%, respectively. While the significant release amounts of propolis from PHBH/30EP started to be detected after 10 min and gradually increased up to 2880 min (48 h) with the release amount of 9.6%. The release of propolis from PHBH/30EP sustained longer than others but the release amount of propolis was less. The fast release of hinokitiol from PHBH/30EH was noticeable in the first 5 min and gradually increased up to 20 min with a maximum release of 46.1%. Afterwards, the release of hinokitiol gradually declined over 240 min (4 h). This decline might be due to the degradation of hinokitiol in PBS solution by light, heat, or solvent (ethanol). In the study of ciprofloxacin hydrochloride (CpHCl) release from electrospun alginate [42], at around 24% release of total loaded drug, the cross-linking process between CpHCl molecules in the nanofibers happened and affected the release behaviors. This result also suggests the relation between the release capacity and the characteristics of PHBH, which is hydrophobic polymer. Ignatova et al. [43] reported that the release of caffeic acid phenethyl ester (CAPE) in material was influenced by hydrophilic–hydrophobic features of the fibrous mats, the release of CAPE increased markedly when CAPE was incorporated in PVP (hydrophilic) matrix than when incorporated in PHB (hydrophobic) matrix.

Especially for PHBH/30EC (10%) and PHBH/30EP (7%), those have similar release behaviors, when only small amounts of natural products were released to PBS. It can be considered due to the high crystallinity of those PHBH composite nanofibers that affected release characteristics. In comparison, PHBH/30EH (1%) with low crystallinity showed fast and high release amounts. Study of drug release of ampicillin incorporated poly(methyl methacrylate)–nylon6 core/shell nanofibers proved that the increase of released drug, most probably due to decreased crystallinity in the polymer matrix [44].

## 4. Conclusions

The natural antibacterial products (*centella*, propolis, and hinokitiol) loaded into PHBH composite nanofibers were successfully fabricated by the electrospinning process. The fiber diameter and surface morphology of PHBH composite nanofibers with antibacterial reagents can be controlled by different kinds of natural antibacterial products, solvent systems for dissolution, and component concentrations in the PHBH-HFIP polymer solutions. The presence of carboxylic acid and aromatic amide groups in PHBH/*centella*, aromatic ring bands due to flavonoid in PHBH/propolis, and tropolone carbon ring in PHBH/hinokitiol were confirmed by FT-IR. The loading of *centella* and propolis led to an increase in the crystallinity of the PHBH polymer. Furthermore, the loading of *centella* and propolis improved the tensile strength, compared to neat PHBH nanofibers. Hinokitiol and propolis were proved to be potent antibiotics by large inhibition zones against both *E. coli* and *S. aureus*. The release of *centella* and hinokitiol from PHBH nanofibers was fast and finished in 20 min with maximum release of 13.9% (PHBH/30EC), 31% (PHBH/30MC), and 46% (PHBH/30EH). Whereas, the release of propolis was continuously to 48 h and maximal release was 9.5%. These results in our study indicated that natural antibacterial products loaded into PHBH composite nanofibers can improve the mechanical properties (PHBH/30EP and PHBH/30EC) and prove the composite nanofibers antibacterial effects against the gram-negative and gram-positive bacteria (PHBH/30EP and PHBH/30EH), which are important characteristics as wound healing materials. Interestingly, PHBH/30EC (10%) showcased good mechanical properties and PHBH/30MC (10%) exhibited good morphology in nanofiber form, however, it may necessary to increase the concentration of *centella* in *centella* solution (ethanol or methanol) if it is to be used as antibacterial reagents. PHBH/30EH (1%) might be used as drug delivery with rapid release but possesses low mechanical properties. PHBH/30EP (7%) could be used in wound healing with needed mechanical properties and slow release for long-time effects.

## Figures and Tables

**Figure 1 nanomaterials-09-01665-f001:**
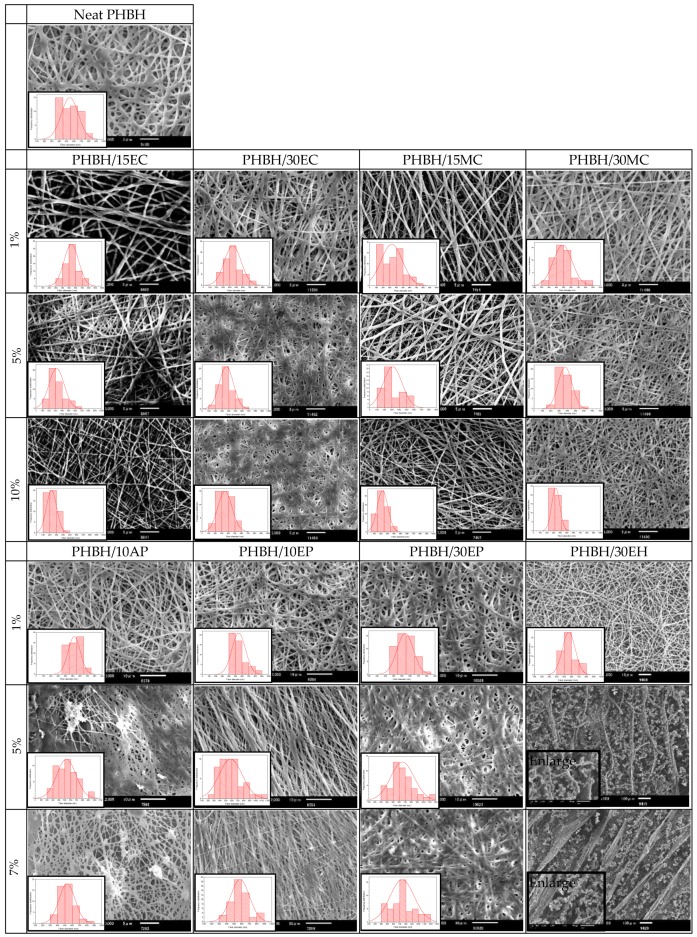
SEM images and diameter distributions of the electrospun neat PHBH 2 wt% and PHBH composite nanofibers containing different natural product solutions. PHBH/15EC, PHBH/30EC, PHBH/15MC, and PHBH/30MC with different concentration ratios of ethanol–*centella* (EC) and methanol–*centella* (MC) (1%, 5%, and 10%). PHBH/10AP, PHBH/10EP, PHBH/30EP, and PHBH/30EH with concentration ratios of 1%, 5%, and 7% of acetone–propolis (AP), ethanol–propolis (EP), and ethanol–hinokitiol (EH).

**Figure 2 nanomaterials-09-01665-f002:**
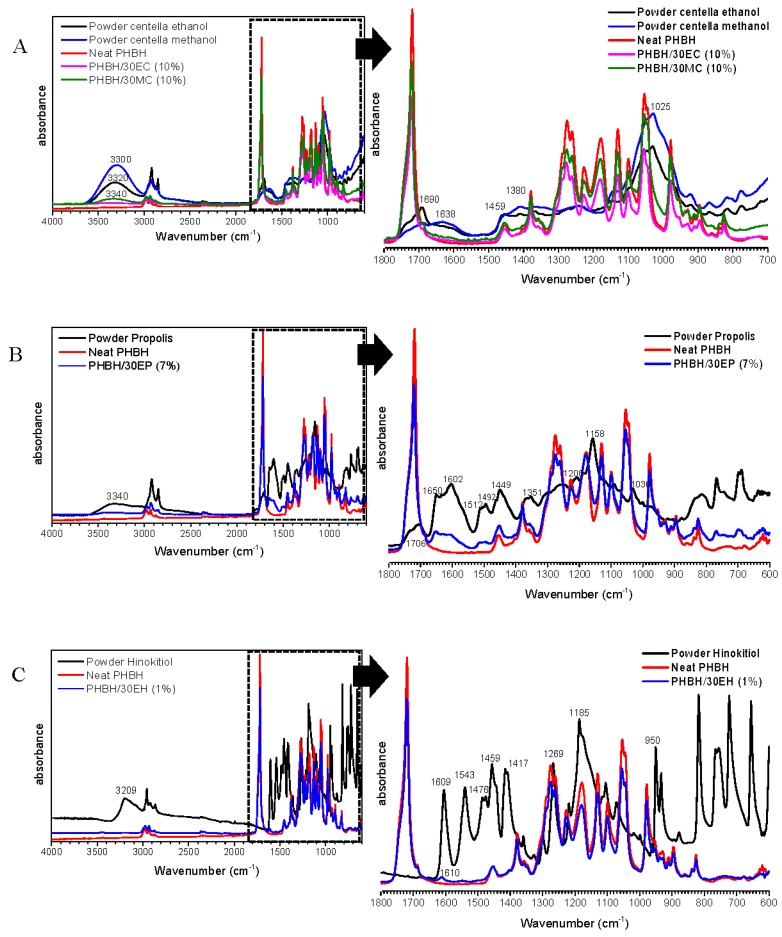
FT-IR spectra of neat PHBH and PHBH composite nanofibers with (**A**) EC and MC, (**B**) EP, (**C**) EH.

**Figure 3 nanomaterials-09-01665-f003:**
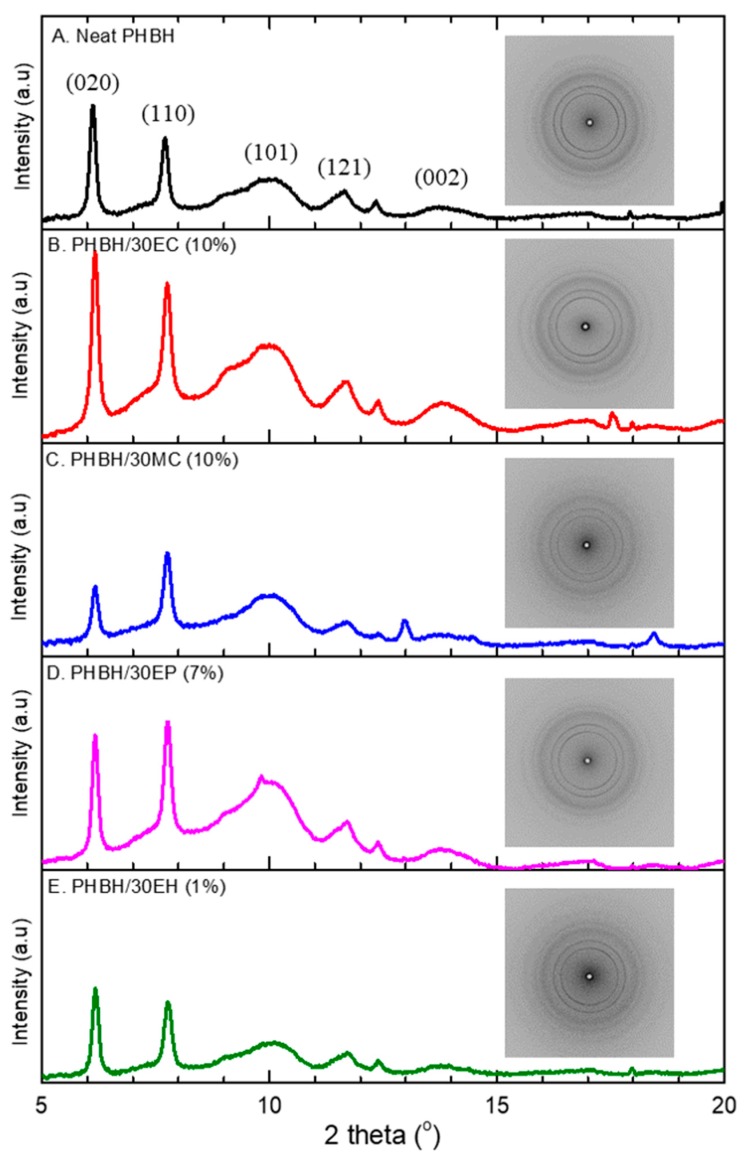
Wide-angle X-ray diffraction (WAXD) intensity profiles and 2D patterns of (**A**) neat PHBH and PHBH composite nanofibers with natural products: (**B**) EC, (**C**) MC, (**D**) EP, (**E**) EH.

**Figure 4 nanomaterials-09-01665-f004:**
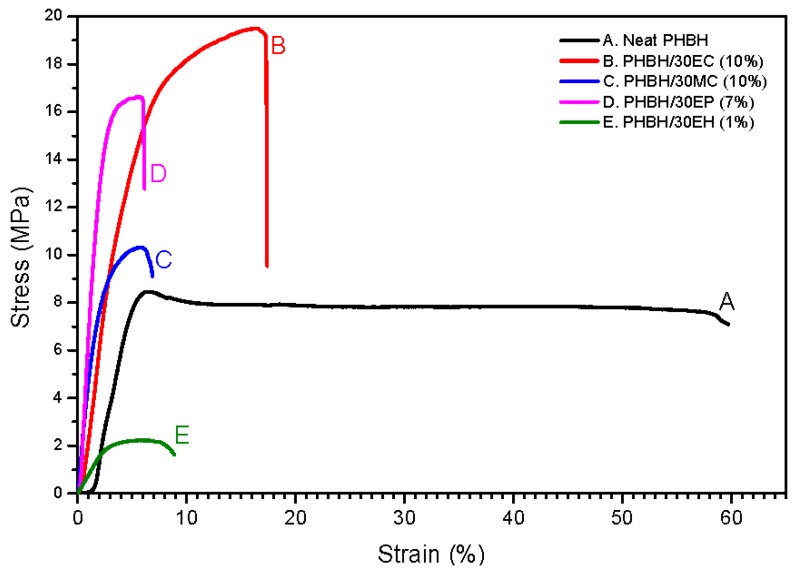
Representative stress-strain curves of (A) neat PHBH and PHBH composite nanofibers with (B) EC, (C) MC, (D) EP, and (E) EH.

**Figure 5 nanomaterials-09-01665-f005:**
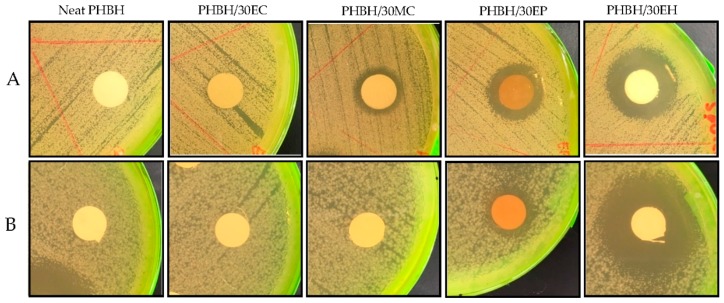
Representative inhibition zones of neat PHBH and PHBH composite nanofibers with *centella* (30EC) and (30MC), propolis (30EP), and hinokitiol (30EH) on gram-positive bacteria (*S. aureus*) (**A**) and gram-negative bacteria (*E. coli*) (**B**).

**Figure 6 nanomaterials-09-01665-f006:**
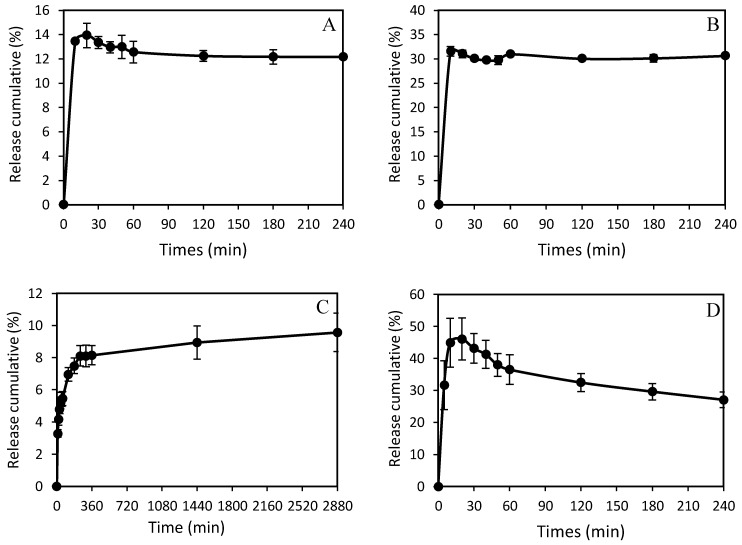
Cumulative release profiles of natural antibacterial products in PHBH composite nanofibers (**A**) PHBH/30EC (10%), (**B**) PHBH/30MC (10%), (**C**) PHBH/30EP (7%), and (**D**) PHBH/30EH (1%).

**Table 1 nanomaterials-09-01665-t001:** The concentrations of natural products in each solution after dry vacuum and ratios of natural product solutions to total weight of spinning poly[(*R*)-3-hydroxybutyrate-*co*-(*R*)-3-hydroxyhexanoate] (PHBH) solutions.

Concentration(*w*/*v*)_Solution_Natural Product (NP)	Percentage Concentration of NP Inside Solution (*w*/*w*)	Ratios NP Solutions over Total Weight of Spinning PHBH Solutions (*v*/*v*) (Code)			
		1%	5%	7%	10%
15% Ethanol–*Centella*	2.16%	15EC (1%)	15EC (5%)	-	15EC (10%)
30% Ethanol–*Centella*	4.14%	30EC (1%)	30EC (5%)	-	30EC (10%)
15% Methanol–*Centella*	3.17%	15MC (1%)	15MC (5%)	-	15MC (10%)
30% Methanol–*Centella*	7.21%	30MC (1%)	30MC (5%)	-	30MC (10%)
10% Acetone–Propolis	8.95%	10AP (1%)	10AP (5%)	10AP (7%)	-
10% Ethanol–Propolis	5.43%	10EP (1%)	10EP (5%)	10EP (7%)	-
30% Ethanol–Propolis	12.25%	30EP (1%)	30EP (5%)	30EP (7%)	-
30% Ethanol–Hinokitiol	30%	30EH (1%)	30EH (5%)	30EH (7%)	-

**Table 2 nanomaterials-09-01665-t002:** Preparation of PHBH with natural product solutions, electrospinning conditions, and fiber diameter of samples.

No.	Polymer	Ratio Concentration-Solvent-Natural Product-Ratio to PHBH Solutions		Electrospinning Conditions		Fiber Diameters		
		Solvent-Natural Product	Ratios to PHBH Solutions (Code)	Applied Voltage (kV)	Flow Rate (mm/min)	Ave. Dia. (nm) ± SD	Max.	Min.
1	Neat PHBH	-	-	20	0.3	543	785	367
2	PHBH	15% Ethanol–*Centella* (EC)	PHBH/15EC (1%)	15.5	0.13	524 ± 111	760	230
			PHBH/15EC (5%)	15.5	0.13	314 ± 116	631	157
			PHBH/15EC (10%)	15.5	0.13	226 ± 77	413	107
3	PHBH	30% Ethanol–*Centella* (EC)	PHBH/30EC (1%)	15.5	0.15–0.2	447 ± 128	789	255
			PHBH/30EC (5%)	15.5	0.15–0.2	362 ± 107	690	131
			PHBH/30EC (10%)	15.5	0.15–0.2	349 ± 108	593	153
4	PHBH	15% Methanol–*Centella* (MC)	PHBH/15MC (1%)	15.5	0.15	319 ± 107	630	152
			PHBH/15MC (5%)	15.5	0.15	310 ± 126	588	135
			PHBH/15MC (10%)	15.5	0.15	227 ± 81	413	101
5	PHBH	30% Methanol–*Centella* (MC)	PHBH/30MC (1%)	15.5	0.15–0.2	487 ± 127	820	238
			PHBH/30MC (5%)	15.5	0.15–0.2	371 ± 83	571	194
			PHBH/30MC (10%)	15.5	0.15–0.2	332 ± 62	456	236
6	PHBH	10% Acetone–Propolis (AP)	PHBH/10AP (1%)	20	0.1	520 ± 83	717	374
			PHBH/10AP (5%)	20	0.1	527 ± 161	978	286
			PHBH/10AP (7%)	20	0.1	529 ± 109	890	328
7	PHBH	10% Ethanol–Propolis (EP)	PHBH/10EP (1%)	20	0.1	548 ± 104	850	422
			PHBH/10EP (5%)	20	0.1	576 ± 193	1192	289
			PHBH/10EP (7%)	20	0.1	579 ± 126	884	338
8	PHBH	30% Ethanol–Propolis (EP)	PHBH/30EP (1%)	20	0.15–0.2	539 ± 99	752	393
			PHBH/30EP (5%)	20	0.15–0.2	653 ± 194	1180	300
			PHBH/30EP (7%)	20	0.15–0.2	739 ± 197	1185	408
9	PHBH	30% Ethanol–Hinokitiol (EH)	PHBH/30EH (1%)	20	0.15–0.2	562 ± 87	745	438
			PHBH/30EH (5%)	20	0.15–0.2	-	-	-
			PHBH/30EH (7%)	20	0.15–0.2	-	-	-

**Table 3 nanomaterials-09-01665-t003:** Mechanical properties and crystallinity of neat PHBH and PHBH composite nanofibers with natural products.

Sample	Tensile Strength	Elongation at Break	Young’s Modulus	Crystallinity
	(MPa)	(%)	(MPa)	(%)
Neat PHBH	8.00 ± 0.71	61.49 ± 17.38	291.5 ± 41.8	47.0
PHBH/30EC (10%)	17.79 ± 4.71 *	15.57 ± 4.56 *	420.4 ± 146.1 *	55.0
PHBH/30MC (10%)	8.98 ± 1.47	10 ± 3.79 *	291.7 ± 118.9	49.2
PHBH/30EP (7%)	16.35 ± 1.78 *	9.27 ± 3.32 *	545.6 ± 162.8 *	54.5
PHBH/30EH (1%)	2.14 ± 0.70 *	8.33 ± 1.75 *	78 ± 33.8 *	44.1

* *p* < 0.05 considered statistically significant between PHBH composite nanofibers in each group against neat PHBH nanofibers (*n* = 10) (except crystallinity, *n* = 2).

**Table 4 nanomaterials-09-01665-t004:** Zones of inhibition for PHBH composite nanofibers with different natural products against *S. aureus* and *E. coli*.

Sample	Inhibition Zones (mm)	
	*S. aureus*	*E. coli*
Neat PHBH	0	0
PHBH/30EC (10%)	0	0
PHBH/30MC (10%)	7.7 ± 6.7	0
PHBH/30EP (7%)	18.3 ± 1.5 *	17.3 ± 5.1 *
PHBH/30EH (1%)	25.7 ± 4.0 *	29.7 ± 1.5 *

* *p* < 0.05 considered statistically significant, PHBH composite nanofibers in each group were compared with PHBH nanofibers (*n* = 3).

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
