# Peer review of "Natural Antibacterial Reagents (Centella, Propolis, and Hinokitiol) Loaded into Poly[(R)-3-hydroxybutyrate-co-(R)-3-hydroxyhexanoate] Composite Nanofibers for Biomedical Applications"

_nanomaterials, 2019, doi:10.3390/nano9121665_

Round 1
Reviewer 1 Report
Manuscript ID: nanomaterials-633071
Title: Natural antibacterial reagents (Centella, Propolis, and Hinokitiol) loaded into poly[(R)-3-hydroxybutyrate-co-(R)-3-hydroxyhexanoate] composite nanofibers for biomedical applications
The language is often difficult to understand, and thus I request the manuscript be thoroughly revised to improve clarity.
In the Introduction section, please state if the cited studies were performed in vitro or in vivo. It has been mentioned for some but not all cited work. And, in case of in vivo studies, please cite if they were performed on humans or other organisms.
I do not understand why the authors constantly refer to the powder content (10% EC, 10% EP, etc.) when defining samples, even though they mention that not all of the powders dissolved in the solvents, and that residues needed to be filtered out. I suggest that the samples be defined by the actual dissolved content of ingredients (they measured it by weighing the residue after evaporation of the solvent). I also suggest that all samples be listed in a table to improve clarity.
“The ratio of ethanol-centella (EC), methanol-centella (EC) were 1%, 5%, and 10% (v/v) for total weight of PHBH solution”. It is difficult to understand this description of ingredient content. A v/v unit is mentioned in connection to total weight.
Was it not necessary to filter out any residues when the powders were dissolved in ethanol / PBS solutions to construct the calibration curves?
The authors mention a 15EP mix on line 247, but I could not find mention of it in the experimental section.
The authors ascribe differences in diameters between neat and ingredient loaded fibers to the effect of solvents on polymer solution viscosity, and the effect of ingredients on polymer solution conductivity, but the chain of argument is not always clear. For example, 30EC (10%) and 30 MC (10%) reduced fiber diameters and that is stated to be due to the effect of ethanol and methanol on polymer solution conductivity, but it is also stated that increasing the percent of EC and MC may also have increased polymer solution conductivity.
I suggest to confirm if ethanol and methanol, when added alone to the polymer solutions, change fiber diameters.
The FTIR spectra in section 3.2 are difficult to read. I suggest that only the section from 1800-600 cm-1 be shown as most differences cited in the paper seem to appear in that region. Also, please consider staggering the graphs so as to improve clarity of understanding.
The last line of section 3.2 reads “The comparison between neat PHBH and PHBH composite nanofibers with natural product revealed no significant difference….” but a whole list of differences are discussed at great length in the section. How significant is that discussion if the differences were not very evident?
Please state how many repetitions of crystallinity determinations were performed, and if possible indicate standard deviation values. Were no statistics performed for Xc values or were there no significant differences?
“As the other reasons, it might be due to the dispersion of natural product in PHBH composite nanofibers and surface state of natural product as well.” This phrase on lines 463-464 is very vague. It appears the authors want to cite all possible reasons for differences between fibers without explaining how precisely these factors affect the properties.
Section 3.3 is difficult to understand.
The * values indication in Table 3 are a puzzle. Was the inhibition zone of 30MC (10%) not different from that of neat fibers?
Author Response
Dear Editor,
Thank you very much for the opportunity to revise and resubmit our manuscript. Constructive comments and suggestions by the reviewers are really appreciated. We have followed the reviewer’s suggestions and provided the additional information in the table and graph of data and results. We respond to the individual remarks and revised the manuscript. Please see the attachment.

Reviewer 2 Report
The paper "Natural antibacterial reagents (Centella, Propolis, and Hinokitiol) loaded into poly[(R)-3-hydroxybutyrate-co-(R)-3-hydroxyhexanoate] composite nanofibers for biomedical applications" concerns polymeric (PHBH) fibers loaded with antibacterials derived form natural sources.
The article is well-written, the experiments are well-planned, and the results presented and described properly. What is very important, the results are discussed with current available literature, what is now rarely done by researchers.
The results are consistent and presented in a final way that facilitates interpretation and proper reception of the purposefulness of conducted research.
I suggest improving the quality of IR spectra shown in Figure 2 to ensure better magnification of presented inserts.
As well, I suggest proofreading because some sentences are incomprehensible.
Finally, I suggest improvment of section concerning the release study, since some of the author's interpretention of obtained results are illogical e.g. ".. the release of hinokitiol was gradually declined below 30% in 240 minutes (4 h) and it might due to the degradation of hinokitiol in PBS solution." How is possible that organic compound (hinokitiol) can degradade in pure PBS. These results are interesting and important, however need more comments. For instance, please concider papers of Gamez et al. "Antimicrobial Electrospun Polycaprolactone-Based Wound Dressings: An In Vitro Study About the Importance of the Direct Contact to Elicit Bactericidal Activity" or of Kyziol et al. " Preparation and characterization of electrospun alginate
nanofibers loaded with ciprofloxacin hydrochloride" since they results on releasing of natural compunds or other antibacterials are similar to yours, .
I strongly recommend this paper to publication in Nanomaterials. I think that reporting on novel biomaterials composed of natural dervatives are exctremely important, among other things from the point of view of looking for new materials to fight increasing antibacterial resistance.
Author Response

(The authors gave the same response as above.)

Reviewer 3 Report
The authors promote the use of natural antibacterial products for medical applications. The work is very interesting and well done, although an extensive English revision is required before considered publication.
Author Response
Dear Editor,
Thank you very much for the opportunity to revise and resubmit our manuscript. Constructive comments and suggestions by the reviewers are really appreciated. We have followed the reviewer’s suggestions and provided the additional information in the table and graph of data and results. We respond to the individual remarks and revised the manuscript.
Reviewer #3:
Comments and Suggestions:
The authors promote the use of natural antibacterial products for medical applications. The work is very interesting and well done, although an extensive English revision is required before considered publication.
Answer:
Thank you very much for your suggestions, we have also revised the grammatical errors in the manuscript.
Round 2
Reviewer 1 Report
Thank you for the corrections. I recommend the paper be accepted for publication.
Reviewer 3 Report
I personally like the modifications introduced in the manuscript.